# Pharmacological Modulation of Energy and Metabolic Pathways Protects Hearing in the Fus1/Tusc2 Knockout Model of Mitochondrial Dysfunction and Oxidative Stress

**DOI:** 10.3390/antiox12061225

**Published:** 2023-06-06

**Authors:** Winston J. T. Tan, Joseph Santos-Sacchi, Jane Tonello, Anil Shanker, Alla V. Ivanova

**Affiliations:** 1Department of Physiology, Faculty of Medical and Health Sciences, The University of Auckland, Auckland 1023, New Zealand; winston.tan@auckland.ac.nz; 2Department of Surgery (Otolaryngology), Yale University School of Medicine, New Haven, CT 06510, USA; joseph.santos-sacchi@yale.edu; 3School of Medicine, Meharry Medical College, Nashville, TN 37208, USA; jtonello@mmc.edu (J.T.); ashanker@mmc.edu (A.S.); 4School of Graduate Studies and Research, Meharry Medical College, Nashville, TN 37208, USA

**Keywords:** age-related hearing loss, cochlea, Fus1/Tusc2, mitochondria, mitochondrial dysfunction, oxidative stress, rapamycin, 2-deoxy-D-glucose, 2-DG, Seahorse analysis

## Abstract

Tightly regulated and robust mitochondrial activities are critical for normal hearing. Previously, we demonstrated that Fus1/Tusc2 KO mice with mitochondrial dysfunction exhibit premature hearing loss. Molecular analysis of the cochlea revealed hyperactivation of the mTOR pathway, oxidative stress, and altered mitochondrial morphology and quantity, suggesting compromised energy sensing and production. Here, we investigated whether the pharmacological modulation of metabolic pathways using rapamycin (RAPA) or 2-deoxy-D-glucose (2-DG) supplementation can protect against hearing loss in female Fus1 KO mice. Additionally, we aimed to identify mitochondria- and Fus1/Tusc2-dependent molecular pathways and processes critical for hearing. We found that inhibiting mTOR or activating alternative mitochondrial energetic pathways to glycolysis protected hearing in the mice. Comparative gene expression analysis revealed the dysregulation of critical biological processes in the KO cochlea, including mitochondrial metabolism, neural and immune responses, and the cochlear hypothalamic–pituitary–adrenal axis signaling system. RAPA and 2-DG mostly normalized these processes, although some genes showed a drug-specific response or no response at all. Interestingly, both drugs resulted in a pronounced upregulation of critical hearing-related genes not altered in the non-treated KO cochlea, including cytoskeletal and motor proteins and calcium-linked transporters and voltage-gated channels. These findings suggest that the pharmacological modulation of mitochondrial metabolism and bioenergetics may restore and activate processes critical for hearing, thereby protecting against hearing loss.

## 1. Introduction

Hearing loss is the most common sensory disability. It is estimated that approximately 430 million people globally are affected by disabling hearing loss, and this is projected to increase to 700 million by 2050 [1]. The most prevalent form of hearing loss is age-related hearing loss (ARHL), or presbycusis, which affects over 65% of individuals above 60 years old [2]. ARHL manifests as a progressive decline in hearing sensitivity with advancing age and a reduced ability to understand speech, particularly in noisy environments. Unaddressed hearing loss can lead to social isolation, loneliness, and depression. Therefore, there is a crucial need to develop novel pharmacological therapies that can prevent or rescue hearing loss.

Emerging evidence suggests that mitochondrial dysfunction and oxidative stress in the cochlea, the peripheral organ of hearing, plays a central role in the development of ARHL as well as other forms of sensorineural hearing loss, such as noise- and ototoxic-drug-induced hearing loss [3,4,5,6,7]. Mitochondrial dysfunction is also implicated in the normal aging process and various age-related pathologies, including neurodegenerative and cardiovascular diseases [8,9]. Mitochondria have vital roles in multiple cellular processes, including energy production, calcium signaling, and apoptosis. Moreover, mitochondria serve as the major intracellular source of reactive oxygen species (ROS), which are by-products of oxidative phosphorylation [10]. The cochlea, particularly the sensory hair cells, stria vascularis, and spiral ganglion neurons (SGNs), has high metabolic demands and is therefore especially vulnerable to the damaging effects of ROS-induced oxidative stress. Excessive mitochondrial production of ROS in cochlear cells can inflict oxidative damage on mitochondrial components, such as mitochondrial DNA (mtDNA mutations), mitochondrial membranes (lipid peroxidation), and respiratory chain proteins (protein oxidation), leading to mitochondrial dysfunction [3,4,5,6]. Decreased mitochondrial function can facilitate further mitochondrial ROS production, ultimately leading to the activation of apoptosis in cochlear cells. This constitutes the basis of the mitochondrial free radical theory of aging [11,12].

We previously established a novel mouse model of early onset ARHL driven by mitochondrial dysfunction and oxidative stress due to deficiency in Fus1, a nuclear-encoded and ubiquitously expressed mitochondrial protein that plays an essential role in maintaining mitochondrial homeostasis [13]. These mice had severe vascular deterioration and atrophy of the stria vascularis, SGN degeneration, reduced inner hair cell (IHC) afferent synapses, loss of type IV fibrocytes in the spiral ligament, and chronic inflammation, which manifested early and progressed with age. Abnormal mitochondria were observed in strial cells and SGNs. In agreement with the severe strial pathology was a markedly reduced endocochlear potential, the driving force for sensory transduction. Moreover, pathological alterations in antioxidant machinery, autophagy, and nutrient- and energy-sensing pathways (mTOR and PTEN/AKT) were detected in the cochlea. These pathologies recapitulate that observed in human metabolic (strial) presbycusis, which is the least characterized type of ARHL. Importantly, prophylactic treatment with the antioxidant N-acetyl-L-cysteine (NAC) prevented ARHL and the associated molecular and pathological changes, strongly indicating the role of oxidative stress as the major causative mechanism. Therefore, this mouse model is a clinically relevant and valuable tool for studying ARHL of primarily metabolic origin linked to mitochondrial dysfunction and oxidative stress.

In this follow-up study, we sought to explore the potential therapeutic benefits of rapamycin (RAPA) and 2-deoxy-D-glucose (2-DG), two clinically approved drugs that modulate molecular pathways involved in energy sensing and production, in Fus1 KO mice. RAPA is a specific inhibitor of the mammalian or mechanistic target of rapamycin (mTOR) signaling pathway, a major nutrient-sensitive regulator of growth and metabolism through its modulation of critical cellular processes, including protein synthesis, autophagy, and apoptosis [14,15,16]. RAPA has been demonstrated to slow down aging and extend the lifespans of animals [17,18,19]. 2-DG, on the other hand, is a synthetic glucose analog that competitively inhibits glycolysis and glucose metabolism and induces a compensatory increase in alternative energetic pathways, such as fatty acid oxidation or glutamine pathways [20,21]. Both drugs also have potential to reduce oxidative stress via increasing the expression of the oxidative stress response genes, removing damaged mitochondria, modulating energy production pathways, and other mechanisms [21,22,23,24,25,26].

Here, we evaluated the therapeutic efficacy of a 3-month oral supplementation of RAPA and 2-DG in protecting against hearing loss in female Fus1 KO mice as compared to wild-type (WT) mice. We also analyzed the downstream molecular pathways altered by these drugs in the cochlea and characterized several classes of Fus1-dependent biological pathways that were improved by or resistant to the RAPA and 2-DG treatments. Furthermore, we used bone-marrow-derived macrophages (BMDMs) as a model to explore the effects of RAPA and 2-DG on the energetics of cochlear macrophages, which are prominently activated in the Fus1 KO cochlea [13]. Altogether, the present work confirms the critical role of mitochondrial metabolic dysfunction in premature hearing loss in Fus1 KO mice and identifies potential genes and pathways that can be targeted for the treatment of ARHL linked to altered mitochondrial metabolism and energy production in the cochlea.

## 2. Materials and Methods

### 2.1. Animals

Female Fus1 knockout (KO) mice and wild-type (WT) mice, aged 4 and 7 months, were used in this study. Fus1 KO mice generated by Dr. A. Ivanova [27] were backcrossed to a 129/Sv background in the laboratory of Dr. S. Anderson (NCI-Frederick). All animal experiments were performed according to a protocol approved by the Yale University Institutional Animal Care and Use Committee (IACUC), and animals were cared for in accordance with the recommendations in the “Guide for the Care and Use of Laboratory Animals” (National Institutes of Health, Bethesda, MD, USA). Both WT and KO mice were raised in the same room within the Yale Animal Resources Center (YARC, (New Haven, CT, USA)). Mice were housed in standard cages and maintained on a 12 h light–dark cycle. They had ad libitum access to drinking water and a normal diet throughout the experiment.

### 2.2. Drug Treatment

Rapamycin (RAPA) and 2-deoxy-D-glucose (2-DG) were orally administered to 4-month-old female Fus1 KO mice ad libitum for 3 months as a supplement in drinking water. To slowly acclimate the mice to the taste, 50 nM RAPA and 2.5 mM 2-DG were administered for the initial week, and then 100 nM RAPA and 5 mM 2-DG were used for the remainder of the treatment period. These doses were based on previous studies [28,29]. The drug water was changed twice a week. Age- and sex-matched control mice were provided with drug-free water. The body weight, appearance, and behavior of the mice were continually monitored throughout the study.

### 2.3. Auditory Brainstem Response (ABR) Measurement

The hearing sensitivity of the mice was measured using auditory brainstem responses (ABRs), a routinely used non-invasive hearing test. ABRs represent synchronized electrical activity in the auditory nerve and ascending central auditory pathways in response to acoustic stimuli. Measurements were carried out within a sound-attenuating booth (Industrial Acoustics Corp., Bronx, NY, USA). The mice were anaesthetized with chloral hydrate (480 mg/kg), injected intraperitoneally (IP), and placed onto a heating blanket (Harvard Animal Blanket Control Unit, Harvard Apparatus Ltd., Kent, England) to maintain their body temperature at 37 °C. Prior to measurement, an otoscopic examination of the tympanic membranes was performed on the anesthetized mouse using an ENT surgical microscope (ZEISS OPMI 1-FC, Carl Zeiss, Oberkochen, Germany) for evidence of otitis media (middle ear infection). Mice displaying signs of otitis media in either or both ears were excluded from the study. The acoustic stimuli for ABRs were produced, and the responses were recorded using a TDT System 3 (Tucker-Davis Technologies, Inc., Alachua, FL, USA) controlled by BioSigRP (TDT), a digital signal-processing software.

ABRs were measured as previously described [13,30,31] by placing subdermal needle electrodes (LifeSync Neuro, Coral Springs, FL, USA) at the vertex (active, noninverting), the right infra-auricular mastoid region (reference, inverting), and the left neck region (ground). The ABRs were elicited with pure-tone pips presented free-field via a speaker (EC1 Electrostatic Speaker, TDT) positioned 10 cm from the vertex. The symmetrically shaped tone bursts were 3 ms long (1 ms raised cosine on/off ramps and 1 ms plateau) and were delivered at a rate of 21 per second. Stimuli were presented at frequencies between 2 and 32 kHz in half-octave steps and in 5 decibel (dB) decrements in sound intensity from 90 dB SPL (sound pressure level) (or 110 dB SPL if thresholds exceeded 90 dB SPL). Differentially recorded scalp potentials were bandpass filtered between 0.05 and 3 kHz over a 15 ms epoch. A total of 400 responses were averaged for each waveform for each stimulus condition. The ABR threshold was defined as the lowest sound intensity capable of evoking a reproducible, visually detectable response. Suprathreshold amplitudes (µV) and latencies (ms) of the initial four ABR waves (waves I, II, III, and IV) were then determined at 16 kHz. The most sensitive frequency range of hearing in mice is 11.3 to 22.6 kHz, and 16 kHz is the half-octave in-between and was therefore chosen for analysis. The analysis was carried out offline in BioSigRP on traces with visible peaks by setting cursors at the maxima and minima (trough) of the peaks. Latency was determined as the time from the onset of the stimulus to the peak, while amplitude was measured by taking the mean of the ∆V of the upward and downward slopes of the peak.

### 2.4. Gene Expression Analysis

Gene microarray analysis was performed to determine the genes that were upregulated and downregulated in the WT and Fus1 KO mouse cochlea following drug treatment. After euthanizing the mice with chloral hydrate (150 mg/kg, IP), the cochleae were extracted from the temporal bones and carefully trimmed to remove any surrounding non-cochlear tissue. The cochleae were stored in RNAlater™ RNA Stabilization Reagent (Qiagen, Hilden, Germany) and frozen at −80 °C to stabilize and protect the RNA. RNAlater™-stabilized frozen cochlear tissues were then homogenized in TRI Reagent (Zymo Research Corp., Irvine, CA, USA), and the RNA was isolated using the Direct-zol™ RNA MiniPrep kit (Zymo Research Corp., Irvine, CA, USA). RNA samples from four cochleae were pooled together for each experimental group.

Gene expression was measured using an Affymetrix GeneChip system, a commercial microarray platform. Analysis was performed using Genecodis software. Then, 1000 upregulated and downregulated genes in the Fus1 KO mouse cochlea, as compared with the WT mouse cochlea, were uploaded to Genecodis and analyzed for pathway enrichment (KEGG analysis). Only significantly enriched pathways (adjusted *p* value is ≤0.05) were presented.

### 2.5. Cells and In Vitro Cell Stimulation

For the mouse bone-marrow-derived macrophage (BMDM) preparation, bone marrow cells were flushed from the femur and tibia bones of Fus1 KO mice between 6 and 12 weeks of age and were grown at 37 °C in a humidified incubator in RPMI-1640 medium containing L-glutamine, 10% fetal calf serum (FCS), and 30% L929 supernatant (i.e., BMDM growth media) for 7–8 days. The differentiated BMDMs were re-plated into Seahorse 96-well tissue culture plates in RPMI-1640 medium containing L-glutamine, 10% FCS, and 20% L929 supernatant 16–20 h prior to cell stimulation. To stimulate the macrophages, the cells were incubated with 100 ng/mL of lipopolysaccharides (LPS) alone in the presence of rapamycin (100 nM) or 2-DG (1 mM) for 5 h.

### 2.6. Seahorse Metabolic Analysis

Analysis of the extracellular acidification rate (ECAR) and oxygen consumption rate (OCR) was performed with a Seahorse XF96 Extracellular Flux Analyzer instrument (Agilent Technologies, Inc., Santa Clara, CA, USA) in BMDMs as a measure of lactate production (a surrogate for the glycolytic rate) and OXPHOS (mitochondrial respiration), respectively. In brief, WT and Fus1 KO BMDMs were seeded overnight in sextuplicate at a density of 1 × 10^5^ cells per well on a Seahorse cell culture plate in RPMI-1640 medium containing L-glutamine, 10% FCS, and 20% L929 supernatant and stimulated with 100 ng/mL LPS. For the Mitostress assay that measures OCR, prior to starting the assay, the cells were washed and incubated in Seahorse Assay Medium (Agilent Technologies, Inc.) supplemented with 1 mM sodium pyruvate, 2 mM L-glutamine, and 25 mM Glucose in a 37 °C incubator without CO_2_ for 45 min. Oligomycin (ATPase inhibitor, 1 µM), FCCP (1 µM), and rotenone/antimycin (0.5 µM each) were injected where indicated, and the ECAR (mpH/min) and OCR (pMoles O_2_/min) were measured in real time. For the glycolytic test assay that measures the acidification rate (ECAR) as a surrogate for the glycolytic rate, BMDMs cultured/activated as stated above were incubated in Seahorse Assay Medium (Agilent Technologies, Inc.) supplemented with 1 mM L-glutamine in a 37 °C incubator without CO_2_ for 45 min. Glucose (25 mM), oligomycin (ATPase inhibitor, 1 µM), and 2-DG (1 mM) were injected where indicated, and the ECAR (mpH/min) and OCR (pMoles O_2_/min) were measured in real time. All mitochondrial parameters were calculated using Wave software (Agilent Technologies, Inc.).

## 3. Results

### 3.1. Fus1 KO Mice Exhibit Significantly Reduced Hearing from 4 to 7 Months of Age

Our previous study demonstrated that Fus1 KO mice develop early onset progressive hearing loss, as revealed by substantial ABR threshold elevations starting at 4 months of age and severe hearing impairment at 12–13 months of age [13]. In the current study, we aimed to establish whether the pharmacological correction of compromised molecular pathways in the Fus1 KO mouse cochlea could ameliorate the progressive hearing loss. Prior to treatment, we performed ABR measurements to establish the baseline hearing thresholds of 4- and 7-month-old female Fus1 KO mice, the ages selected for the beginning and end of the treatments (Figure 1). Because previous studies have demonstrated sex-specific differences in the efficacy of certain treatments [32,33,34], we decided to use only female mice to limit potential confounding factors. The hearing thresholds of age-matched WT mice were also measured for comparison. The 4-month-old Fus1 KO mice had moderately elevated (15–20 dB SPL) thresholds at the middle and high frequencies as compared to the age-matched WT mice (Figure 1B). At 2 and 11.3 kHz, there was no difference between the WT and Fus1 KO mice (Figure 1B). The 7-month-old Fus1 KO mice had substantial threshold elevations across all frequencies tested as compared to 7-month-old WT mice (Figure 1C), which showed only slight threshold elevation as compared to 4-month-old WT mice (Figure 1A). Thus, the systemic loss of Fus1 affects auditory function across most sound frequencies tested, starting at 4 months of age, and results in significant hearing loss by 7 months of age (Figure 1D).

### 3.2. Supplementation with Drugs Targeting Energy-Sensing and -Producing Pathways Significantly Delays Hearing Loss in Fus1 KO Mice

We previously demonstrated hyper-activation of the mTOR signaling pathway in the cochleae of Fus1 KO mice starting from 3 months of age [13]. We also observed that aerobic glycolysis (mitochondrial respiration) is compromised in Fus1 KO cells [35]. Here, based on the hypothesis that deficiency in energy production, balance, and metabolism may underlie premature hearing loss, we used rapamycin (RAPA), an mTOR inhibitor [14,15,16], and 2-deoxy-D-glucose (2-DG), a non-metabolizable glucose analog that inhibits glycolysis and activates alternative metabolic pathways [20,21].

After the exclusion of the study of animals with significantly impaired hearing, we formed three groups of female Fus1 KO mice: control (n = 8), RAPA (n = 9), and 2-DG (n = 8). The RAPA and 2-DG groups received RAPA and 2-DG, respectively, as a supplement in drinking water for 3 months. All groups showed no signs of distress for the duration of the treatment; however, two mice from the 2-DG-treated group died close to the end of the study. While we cannot definitively attribute the deaths of these mice to 2-DG toxicity, we acknowledge the possibility that it may have played a role. At the end of the treatment, the hearing sensitivity of the 7-month-old KO mice was evaluated using ABRs, and each group was compared with the non-treated control group of 4-month-old Fus1 KO mice (n = 24). We showed that RAPA treatment prevented or delayed the progression of hearing loss in the Fus1 KO mice across the frequencies tested (Figure 1E). Oral administration of 2-DG also showed a strong protective effect against hearing loss in the Fus1 KO mice but only in the 2–11 kHz frequency range (Figure 1F).

### 3.3. Reduced ABR Wave Latencies in Rapamycin-Treated Fus1 KO Mice

To further characterize the ABR responses, we performed an analysis of the input/output (I/O) functions of ABR waves I to IV. The amplitude of the ABR waves reflects the number of activated neurons and synchrony of firing, while latency, the elapsed time from sound delivery, reflects the timing of synaptic transmission and nerve conduction. The amplitude and latency analysis of ABR waves provides information on the integrity of auditory periphery and brainstem pathways.

A comparison of ABR wave I amplitudes and latencies at 16 kHz between the control and treated Fus1 KO mice is shown in Figure 1G–L. The ABR wave II–IV amplitudes and latencies showed a similar pattern to ABR wave I and are shown in Appendix A, respectively. The amplitudes of waves I–IV did not significantly differ between the treated and untreated 7-month-old-mice at any sound intensity (Figure 1G–I and Appendix A). However, we observed a clear protective effect of RAPA in reducing wave I–IV latencies (Figure 1K and Appendix A). Paradoxically, in the 2-DG-treated group, wave I–III latencies at moderate sound intensities (50–75 dB SPL) were profoundly delayed, while at higher sound intensities (80–110 dB SPL), no changes were observed between the untreated and 2-DG-treated mice (Figure 1L and Appendix A). Interestingly, wave IV latencies at high sound levels (105–110 dB SPL) were shortened in the 2-DG-treated mice (Appendix A).

Thus, our analysis of the ABR waveforms reveals that RAPA and 2-DG affects the integrity of the peripheral and central auditory pathways, specifically in relation to ABR wave latencies.

### 3.4. Gene Expression Analysis Reveals Fus1-Dependent Changes in the Immune, Neuronal, and Metabolic Components of the Cochlea

To provide insight into the molecular processes with Fus1 involvement, we isolated RNA from the cochleae of 7-month-old female Fus1 KO and WT mice and performed comparative gene expression analysis. To consider interindividual heterogeneity, RNA specimens pooled from three mice in each group were used. Overall, we identified 1061 genes upregulated and 982 genes downregulated at least 1.3-fold in the Fus1 KO mouse cochlea with a low expression level threshold of 4.0 in the log2 scale (Appendix A).

We performed KEGG (Kyoto Encyclopedia of Genes and Genomes) module GSEA (Gene Set Enrichment Analysis) analysis of the expression data using the GeneCodis portal (http://genecodis.cnb.csic.es/, accessed on 1 March 2023). The KEGG pathways that showed statistically significant gene enrichment in the Fus1 KO cochlea (*p* < 0.01) are listed in Appendix A. “Neuroactive ligand–receptor interaction” was the most enriched KEGG pathway affected by Fus1 loss (Appendix A). Corrected hypergeometric *p* values were 1.7 × 10^−9^ for upregulated genes and 6 × 10^−3^ for downregulated genes.

To better understand the biological significance of Fus1 deletion in the cochlea, we grouped all the enriched KEGG pathways into several categories based on their involvement in different biological processes (Table 1 and Appendix A). Overall, enriched KEGG pathways associated with the immune system and metabolism were the most well-represented. Thus, 17 different immune KEGG pathways were downregulated (Table 1), suggesting the critical role of immune protection in the cochlea. On the other hand, biological processes linked to metabolism were upregulated in the Fus1 KO cochlea (nine KEGG pathways) (Table 1). These nine metabolic pathways included the metabolism of lipids and fatty acids, glycerolipids, sugars, and diseases associated with their dysregulation (i.e., diabetes). It is worth noting that genes from the “Cytokine–cytokine receptor interaction” pathway were enriched in both the upregulated and downregulated categories in the Fus1 KO mouse cochlea, suggesting that Fus1 is involved in the coordinated regulation of immune responses (Appendix A).

Interestingly, several biological processes, such as cell adhesion, the cell cycle, and signal transduction, were detected only in the downregulated gene set, suggesting a critical role of these processes in normal cochlea function (Appendix A). Table 1 summarizes the dysregulation of Fus1-dependent biological processes in the cochlea that may underlie the early hearing loss observed in the Fus1 KO mice.

### 3.5. Deregulation of the Hypothalamic–Pituitary–Adrenal (HPA) Axis in the Fus1 KO Cochlea

Recently, it was suggested that the cochlea uses an analog of the neuroendocrine system, a local hypothalamic–pituitary–adrenal (HPA)-axis-equivalent signaling system that plays a crucial role in auditory processing [36]. Indeed, since the HPA axis plays a major role in the coordination of homeostatic systems of the body [37,38], it may also regulate the homeostasis of different cochlear components. Remarkably, in agreement with this hypothesis, we identified elements of this system in our expression analysis and found out that the expression of key HPA axis hormones was significantly altered in Fus1 KO cochleae (Appendix A). Thus, Gh (growth hormone); Cga (alpha subunit common for chorionic gonadotropin (CG), luteinizing hormone (LH), follicle-stimulating hormone (FSH), and thyroid-stimulating hormone (TSH)); and Prl (prolactin) were significantly upregulated in the KO mice (2.89-, 2.87-, and 2.56-fold increase, respectively). Meanwhile, four other genes, namely Pomc (pro-opiomelanocortin-alpha, a precursor protein of endorphins, β-lipotropin (β-LPH), the adrenocorticotropic hormone (ACTH), and α-, β-, and γ-melanocyte-stimulating hormone (MSH)), and three regulators of the HPA axis signaling, namely, Tac2 (tachykinin 2), Calca (encodes the peptide hormones calcitonin, calcitonin gene-related peptide and katacalcin), and Spx (Spexin), were downregulated in the KO mice (1.85-, 1.99-, 1.8-, and 1.57-fold, respectively). Interestingly, the cognate receptors for these hormones did not follow this trend, except for Oprm1, a receptor for α- and β-endorphins, and Pomc derivatives (Appendix A), suggesting a selective deregulating effect of Fus1 loss on the expression of genes encoding neuropeptide hormones of the HPA axis.

### 3.6. Rapamycin and 2-DG Treatments Improve Immune and Metabolic Pathways That Play Critical Roles in Hearing and Aging as Revealed by KEGG Analysis

To identify key Fus1- and energy-dependent pathways in the cochlea, we compared gene expression in 7-month-old WT cochleae with 7-month-old untreated, RAPA-treated, and 2-DG-treated Fus1 KO cochleae. Differential gene expression was presented as a fold or percentage change as compared to WT mice. To identify energy- and mitochondria-sensitive pathways in the Fus1 KO cochlea, we used KEGG GSEA analysis, as described above. KEGG analysis of approximately 900–1000 up- or downregulated genes clearly showed enrichments in several biological categories affected by either one or both drugs (Table 2 and Table 3). Interestingly, both the RAPA and 2-DG treatments normalized and further upregulated numerous genes associated with the immune system, with RAPA showing more robust pro-immune effects (Table 2). Thus, in the upregulated gene set, the drug treatments increased the number of enriched immune pathways from 1 to 10 (RAPA) and from 1 to 4 (2-DG), while in the downregulated gene set, the number of “immune” categories decreased from 17 to 0 (RAPA) and from 17 to 8 (2-DG). Thus, both RAPA and 2-DG normalized/activated the expression of immune-related genes.

2-DG selectively normalized (downregulated) the expression of genes involved in metabolism (Table 3). The KEGG analysis of upregulated genes in the KO mice revealed enrichment in only one metabolic pathway in the treated mice as compared to nine metabolic pathways identified in the control mice.

RAPA selectively increased (activated) the expression of genes involved in signal transduction and cell adhesion (Table 2). In addition, RAPA increased the expression of genes from the “calcium signaling and associated diseases” category (Table 2). The expression of this group of genes was not significantly affected in the untreated KO cochleae, suggesting that RAPA may have a beneficial effect on calcium-dependent pathways, at least in the context of Fus1-mediated mitochondrial dysfunction. Another interesting observation was that the top ~1000 genes downregulated in the RAPA-treated group were not enriched in any of the KEGG pathways, unlike the top ~1000 downregulated genes in the control KO group, suggesting a normalizing effect of RAPA on many pathways that were downregulated in the control KO group.

### 3.7. Manual Functional Analysis of the Top 100 Differentially Expressed Genes Corroborates the Results of the KEGG Analysis and Provides Additional Data

To corroborate the results from the KEGG analysis and include genes not fully annotated in this database, we performed a manual analysis of functional annotations to the 100 top-ranked genes differentially expressed in the Fus1 KO cochlea using the GeneCards database (Weizmann Institute, Rehovot, Israel) (Appendix A). The immune category was the largest in this analysis, represented by 37 downregulated and 16 upregulated genes (Figure 2A, Appendix A). Analysis of the drug-induced changes showed consistent normalization (closer to the WT mice) of expression among the downregulated genes and either normalization or further increase in the expression of genes from the upregulated group. The second largest category was represented by a set of metabolic genes (9 downregulated and 29 upregulated), which also showed normalization of expression by both drug treatments in both the up- and downregulated groups (Figure 2B, Appendix A).

The third-largest group of genes/proteins that we identified in the analysis (6 downregulated and 11 upregulated) was represented by membrane transporters and voltage-gated channels (VGC), which are known to be vital for auditory processing [39] (Figure 2C, Appendix A). The RAPA and 2-DG treatments also resulted in the normalization of gene expression in both the up- and downregulated groups.

The next three groups of genes differentially expressed in the Fus1 KO cochlea belonged to categories that are critically important for sound processing: neuroreceptors (eight upregulated, one downregulated), HPA axis signaling proteins (three upregulated, four downregulated), and synaptic/synaptogenic proteins (three upregulated and five downregulated) (Appendix A). The bar graphs in Figure 3 illustrate the expression of these genes in the control KO cochlea and drug-induced changes in their expression.

The expression of nine neuroreceptor genes that were significantly altered in the Fus1 KO cochlea was either partially or fully corrected by one or both drugs (Figure 3A, Appendix A). The most consistent effect observed was that of 2-DG treatment, as it showed a corrective effect for eight of the nine genes (Gabra2, Vipr1, Htr3a, Uts2r, Gpr83, Grin3a, Chrnb1, and Oprm1), while RAPA corrected only four genes (Uts2r, Gpr83, Grin3a, and Chrnb1). 

Most of the HPA axis genes responding robustly to the drug treatments (Figure 3B, Appendix A). Interestingly, genes overexpressed in the control KO cochlea (Prl, Cga, and Gh) showed not only the normalization of expression but also further downregulation in the drug-treated KO mice. Genes downregulated in the control KO mice (Spx, Calca, Tac2) showed a trend towards normalization after treatment with either drug. Paradoxically, the expression of Pomc, an HPA axis gene of paramount importance that encodes multiple neuropeptide hormones, was not corrected. Moreover, RAPA treatment led to the further downregulation of Pomc, suggesting that decreased POMC expression in the KO cochlea is a critical compensatory change caused by the loss of Fus1.

The majority of genes from the synaptic/synaptogenic protein group showed full or partial correction by both drugs, except for Lrrtm4, which responded only to RAPA, and Pcdhb22, which was only corrected by 2-DG (Figure 3C, Appendix A).

### 3.8. Increased Expression of Cytoskeletal and Motor Proteins, Calcium-Linked Transporters and Voltage-Gated Channels (VGCs), and Proteins Involved in Mitochondrial Energy Production in the RAPA and 2-DG-Treated Fus1 KO Cochlea

Analysis of the 100 top-ranked genes highly expressed in the RAPA and 2-DG-treated Fus1 KO cochlea revealed a very strong enrichment (45 out of 100) of cytoskeletal and motor proteins (Figure 4A, Appendix A). Figure 4A and Appendix A show that most of these genes were not significantly altered in the KO cochlea prior to treatment; however, they showed a robust, up to 16-fold increase in expression after treatment. It is worth mentioning that while both the RAPA and 2-DG treatments led to the increased expression of these genes in the KO cochlea, RAPA produced a more robust effect.

Another group of proteins that were upregulated by RAPA and less robustly upregulated by 2-DG, while remaining unaffected in the control KO cochlea, were transporters and voltage-gated channels (VGC) that are functionally associated with calcium-dependent processes (Figure 4B, Appendix A). These data corroborated the KEGG analysis data, which identified five pathways linked to calcium signaling and associated diseases in the top ~1000 genes upregulated by RAPA.

The third group of genes activated by both RAPA and 2-DG in the KO cochlea were proteins involved in energy production and homeostasis (Figure 4C, Appendix A). It is noteworthy that among the 9 proteins, only Cox6a2 is involved in ATP production via OXPHOS, while the rest of the proteins are involved in glycolysis, gluconeogenesis, glycogenolysis, ketogenesis, and chemiosmotic coupling (9 out of 100). This suggest that the modulation of pathways other than OXPHOS energetic pathways have a promising therapeutic application. RAPA activated the expression of these and other proteins more robustly than 2-DG, suggesting the more universal energy-modulating effects of rapamycin treatment. Table 4 is a summary of the biological processes that are upregulated or downregulated in the cochleae of 7-month-old Fus1 KO mice in response to RAPA and 2-DG treatment.

### 3.9. Treatment of Activated Fus1 KO Macrophages with Rapamycin or 2-DG Improves Energy Metabolism of Myeloid Cells

In our previous study, we found that Fus1 KO mice have a higher number of activated cochlear macrophages in the bone marrow of the otic capsule and an increased infiltration of activated macrophages in the spiral ligament, suggesting chronic inflammation in the Fus1 KO cochlea [13]. The activation of immune cells results in profound changes in their energy metabolism [40,41], which, if chronic, can be harmful to immune cells and surrounding tissue functions [42]. Thus, we investigated if and how the energy metabolism of activated Fus1 KO and WT macrophages differ and if it could be modulated using RAPA or 2-DG treatments.

For this experiment, we analyzed the energetic metabolism of bone-marrow-derived macrophages (BMDM) from young (2-month-old) WT and Fus1 KO mice with the Seahorse XFe96 Analyzer (Agilent), which measures the real-time oxygen consumption rate (OCR) (Figure 5A,B) and extracellular acidification rate (ECAR) (Figure 5C) of live cells. These rates are key indicators of mitochondrial respiration and glycolysis, as well as the ATP production rate. We compared the metabolism of LPS-activated macrophages and found a profound difference between Fus1-proficient and Fus1-deficient LPS-activated BMDMs. Most of the mitochondrial respiration parameters (basal and maximal respiration, non-mitochondrial respiration, and spare respiratory capacity) were significantly increased in young Fus1 KO BMDMs, as compared to WT macrophages (Figure 5B). Since eukaryotic cells use oxidative phosphorylation (OXPHOS), glycolysis, and the tricarboxylic acid (TCA) cycle to satisfy their energy requirements, we also tested the ECAR, which represents energy production via anaerobic glycolysis. We also observed a statistically significant difference in ECAR between the WT and KO activated macrophages (Figure 5C).

The short treatment (5 h) of the BMDMs with RAPA and 2-DG revealed that while the statistical difference between the treated WT and KO macrophages was maintained (Figure 5B,C), the comparison of the non-treated WT with RAPA- or 2-DG-treated KO macrophages showed the normalization of several parameters of mitochondrial respiration in the KO BMDMs (basal respiration: WT vs. KO 2-DG *p* = 0.4 compared to WT vs. KO *p* > 0.05; ATP production: WT vs. KO 2-DG *p* = 0.4 compared to WT vs. KO *p* > 0.003) (Figure 5B). We suggest that this increased mitochondrial respiration and glycolysis are a compensatory mechanism in response to the mitochondrial deficiency and oxidative stress caused by the loss of Fus1. Constantly high energy production might be detrimental to tissue metabolism and physiologic functioning, which could lead to the loss of tissue homeostasis and premature aging. Thus, an attractive strategy for preventing or delaying hearing loss would be the modulation of energy-producing and -regulating pathways in the cochlea.

## 4. Discussion

Our previously established mouse model of ARHL formed through the deletion of Fus1, a critical mitochondrial protein, highlights the essential role of mitochondrial dysfunction in the development of ARHL [13]. In the present study, we demonstrated that oral supplementation with RAPA or 2-DG, drugs targeting molecular pathways related to energy sensing and production, delayed the progression of ARHL in female Fus1 KO mice.

ARHL is classified according to the primary temporal bone pathology and audiometric findings, with the three main types being sensory, neural, and metabolic/strial ARHL [43]. Sensory presbycusis primarily involves damage to the sensory hair cells, while neural presbycusis is caused by damage to the spiral ganglion neurons and central auditory pathways, and metabolic or strial presbycusis is due to the atrophy and functional impairment of the stria vascularis. ARHL can also be of mixed pathology, resulting from a combination of more than one type of ARHL. A growing body of evidence indicates that mitochondrial dysfunction and oxidative stress are implicated in all three types of ARHL, as sensory hair cells, spiral ganglion neurons, and strial cells all have high metabolic demands and are thus susceptible to mitochondrial oxidative damage [3,4,5,6,7]. The Fus1 KO mouse model mainly recapitulates the pathophysiology of metabolic presbycusis, the least understood type of ARHL [13]. These mice are characterized by reduced mitochondrial bioenergetic capacity, causing severe strial pathology, including vascular deterioration and atrophy, and a markedly reduced endocochlear potential [13].

Using comparative gene expression analysis, we identified numerous genes and pathways in the cochlea linked to progressive hearing dysfunction in Fus1 KO mice. Among the altered pathways in the cochlea, “neuroactive ligand–receptor interaction” was the most enriched KEGG pathway affected by the loss of Fus1. This finding, along with the premature hearing loss observed in the Fus1 KO mice, strongly suggests that both Fus1 and mitochondria play significant roles in regulating neuronal processes involved in auditory signal transduction and synaptic transmission. This is also corroborated by the markedly reduced amplitudes and prolonged latencies of ABR waves in Fus1 KO mice [13]. Suprathreshold ABR wave amplitudes and latencies provide an objective measure of synaptic and auditory nerve integrity.

KEGG pathways linked to immunity were highly represented in the Fus1 KO mouse cochlea, with 17 pathways being downregulated. This implies that immune protection in the cochlea is Fus1- and mitochondria-dependent and critical for normal cochlear function. Although previously believed to be an immune-privileged organ isolated from the immune system, it is now evident that the cochlea can mount an inflammatory response to various acute and chronic stresses, such as pathogens, foreign antigens, acoustic overstimulation, ototoxic drugs, and aging [44,45,46,47,48]. Cochlear tissues contain resident immune cells (e.g., macrophages) that express various inflammatory mediators and are implicated in the early activation and resolution of the immune response in the cochlea [49,50]. Among the downregulated immune-related genes in the Fus1 KO cochlea were those related to the toll-like receptor signaling pathway. This pathway has previously been shown to be expressed in the supporting cells of the organ of Corti, indicating that resident cells in the cochlea also possess immune capabilities [51].

Furthermore, numerous metabolic-related pathways were highly upregulated, including pathways involved in the metabolism of lipids and fatty acids, glycerolipids, sugars, and diseases associated with their dysregulation (i.e., diabetes). Such consistent upregulation of metabolic pathways in the Fus1 KO cochlea suggests an increased usage of alternative energy-producing pathways, such as anaerobic glycolysis and fatty acid oxidation, to compensate for the reduced bioenergetic capacity caused by mitochondrial dysfunction [13,35]. These biological pathways in Fus1 KO mice provide important insights into the underlying mechanisms of hearing loss and highlight potential targets for future therapies aiming to prevent or treat hearing loss. The limitation of the current study is the lack of further detailed characterization of changes in individual genes critical for hearing, such as the synaptic proteins, bassoon (BSN) and brain-derived neurotrophic factor (BDNF). The future perspective study should address this limitation.

Based on our hypothesis that deficiency in energy production, balance, and metabolism may underlie premature hearing loss in Fus1 KO mice, we explored the therapeutic potential of RAPA and 2-DG for protecting against ARHL. Recently, there has been growing interest in RAPA as a potential intervention to slow down the aging process, based on studies suggesting that it can increase lifespan in certain organisms [17,18,19]. RAPA has been reported to delay the age-related loss of OHCs and ARHL in UM-HET4 mice when administered during early and late midlife, respectively [52,53]. Moreover, the oral administration of RAPA to C57BL/6J mice enhanced autophagy in SGNs, leading to the decreased apoptosis of SGNs and amelioration of ARHL [54]. Our previous findings also revealed hyperactivation of the mTOR pathway and reduced autophagy in the Fus1 KO mouse cochlea [13]. 2-DG, on the other hand, is a synthetic glucose analog that competitively inhibits glycolysis and induces a compensatory increase in alternative energetic pathways, such as the generation of ketone bodies in mitochondria via fatty acid oxidation [20,21]. At present, no previous studies have suggested a protective role of 2-DG against hearing loss.

Our study found that Fus1 KO mice treated with RAPA and 2-DG showed significant improvement in their ABR thresholds. RAPA was effective in preventing the decline in hearing sensitivity across all frequencies, whereas 2-DG only provided protection against threshold elevation in the middle to low frequencies. The preferential effect of 2-DG on lower frequencies is potentially due to the differential levels of antioxidants in basal and apical OHCs, with basal cells being more susceptible to ROS-induced damage due to their lower antioxidant levels [55]. Further investigation is required to identify the precise mechanism behind 2-DG’s effect on hearing thresholds at different frequencies. These findings suggest that the better hearing protection provided by RAPA, compared with 2-DG, is attributed to its influence on a greater number of biological pathways in the cochlea, as discussed below.

In addition to protecting hearing thresholds, RAPA and 2-DG were effective in restoring the expression of genes associated with various biological processes that were altered in the Fus1 KO cochlea, as demonstrated by both KEGG/GSEA analysis and manual functional analysis. Interestingly, both drugs normalized or further upregulated numerous immune-related genes, with RAPA exhibiting stronger pro-immune effects. Therefore, the enhanced immune capacity of the cochlea in response to RAPA and 2-DG treatment may play a crucial role in countering hearing loss in Fus1 KO mice by improving cochlear homeostasis.

The expression of most genes from the synaptic and synaptogenic protein group was fully or partially corrected by both RAPA and 2-DG. Notably, BSN and BDNF, which were downregulated in the Fus1 KO cochlea and corrected by both drugs, have been linked to auditory function, specifically IHC synaptic transmission [56,57]. Bassoon is a large presynaptic scaffold protein that plays an important role in the organization and function of IHC synapses. Bassoon mutant mice have impaired sound encoding due to their reduced functional presynaptic ribbons [56]. BDNF, which has been shown to reduce with age in rats and gerbils [57], is important for normal exocytosis and the maintenance of the ribbon number in IHC synapses [58]. Our findings indicate that the increased expression of BSN and BDNF may underlie the protection against hearing loss in RAPA and 2-DG-treated Fus1 KO mice. This is reflected not only in the improved hearing thresholds but also in the significantly shortened ABR wave latencies caused by RAPA at higher sound intensities. Latency refers to the timing of synaptic transmission and nerve conduction along the auditory pathway, with wave I representing the activities of the hair cell and auditory nerve fibers.

Cytoskeletal and motor proteins were also highly upregulated in the Fus1 KO cochlea as a result of 3-month RAPA and 2-DG supplementation while remaining unaltered in the untreated Fus1 KO mice. Cytoskeletal proteins, such as actin, and motor proteins, such as myosin, are crucial for maintaining the structure and function of cells that require mechanical movement, such as skeletal muscle cells [59]. In the cochlea, they play a vital role in regulating the length and function of stereocilia [60,61], which are hair-like projections on the apical surface of hair cells that are responsible for mechanoelectrical transduction. Additionally, cytoskeletal proteins also help to reorganize the mechanical properties of hair cells. OHCs consist of a cortical cytoskeletal lattice, a highly elaborate and organized structural network of circumferentially arranged actin filaments that are cross-linked by longitudinally arranged spectrin filaments [62,63]. It has been speculated that the OHC cytoskeleton is involved in harnessing forces generated by the voltage-dependent motor protein prestin within the plane of the plasma membrane and directing them along the longitudinal axis of the OHC to effect changes in cell length, termed electromotility [64,65,66]. Prestin-driven electromotility is the cellular basis of cochlear amplification, a process responsible for the sensitivity and frequency selectivity of mammalian hearing [67,68,69,70]. As a result, the enhanced expression of cytoskeletal and motor proteins in stereocilia and hair cells due to RAPA and 2-DG treatment may protect hearing by improving hair cell mechanoelectrical transduction and OHC electromechanical activity.

Another biological pathway that was enhanced by RAPA and, to a lesser extent, by 2-DG was calcium signaling. This pathway was largely unaffected in the untreated KO cochlea, suggesting that RAPA may have a beneficial effect on calcium-dependent signaling pathways in the Fus1 KO cochlea. Mitochondria play a crucial role in regulating intracellular calcium signaling [71]. We previously showed that the loss of Fus1 in cells results in the dysregulation of calcium fluxes to/from mitochondria [35,72,73,74]. Calcium ions regulate multiple aspects of cochlear physiology, including mechanoelectrical transduction, receptor potential modulation, and synaptic transmission in hair cells [75,76]. The increased expression of voltage-gated calcium channels and other proteins that regulate calcium levels may be responsible for the reduced ABR wave I latencies observed in RAPA-treated KO mice due to increased calcium-mediated neurotransmitter release at IHC ribbon synapses. Although 2-DG was also found to increase the expression of several genes related to calcium signaling, it unexpectly caused a further delay in ABR wave latencies at moderate sound intensities, and this phenomenon warrants further investigation.

Our comparative studies of energetic metabolism in activated Fus1 KO and WT BMDMs showed statistically significant elevation of all OXPHOS and glycolytic parameters in the Fus1 KO BMDMs, suggesting that Fus1 is needed to maintain the homeostasis of energetic pathways. Thus, when lost, the chronic activation of mitochondrial and non-mitochondrial energy production pathways will eventually lead to tissue dysfunction due to the increased generation of mitochondrial ROS, reduced mitochondrial membrane potential, mitochondrial calcium imbalance, mtDNA damage, abnormal mitophagy, etc. These dysregulations cause oxidative stress, inflammasome activation, apoptosis, senescence, and metabolic reprogramming [35,72,73,74,77,78,79]. All these cellular processes participate in the pathogenesis and progression of chronic age-related diseases, including hearing loss [80]. Our findings demonstrated that a 5 h treatment with 2-DG in vitro was sufficient to reduce basal mitochondrial respiration and ATP production in Fus1 KO BMDMs to levels comparable to those observed in WT BMDMs. Thus, an optimized protocol of chronic RAPA and 2-DG treatments could stabilize Fus1 KO energetic metabolism, which could potentially normalize critical cellular processes in the cochlea.

Both RAPA and 2-DG are orally available and relatively non-toxic and, thus, are attractive therapeutic options for hearing loss. Because oxidative stress and chronic inflammation (inflammaging) in the cochlea are thought to be key contributing factors in the pathogenesis of age-related cochlear cell degeneration and hearing loss [47,80,81,82,83,84,85,86], exploring the synergistic potential of a combination therapy consisting of both RAPA and 2-DG together with mitochondria-targeted antioxidants and anti-inflammatory drugs to completely prevent or slow down ARHL is a promising area of research with the potential to lead to novel therapeutic approaches. However, further research is needed to determine the ideal combination of drugs, as well as the optimal dosage and timing of treatment, in order to ensure a safe and effective therapy for ARHL in humans.

## 5. Conclusions

In conclusion, our study demonstrates that premature hearing loss driven by mitochondrial dysfunction can be delayed through the pharmacological correction of energy-sensing and -producing pathways in the cochlea. Using comparative gene expression analysis, we found that a 3-month oral supplementation with RAPA or 2-DG can fully or partially correct, or further upregulate, the expression of essential genes associated with various biological processes that are altered in the Fus1 KO cochlea, including immunity, metabolism, synapses, cytoskeleton/motor proteins, and calcium signaling. Collectively, our findings not only provide crucial insights into the underlying mechanisms of metabolic ARHL but also pave the way for novel approaches to protect against hearing loss by restoring mitochondrial bioenergetics.

## Figures and Tables

**Figure 1 antioxidants-12-01225-f001:**
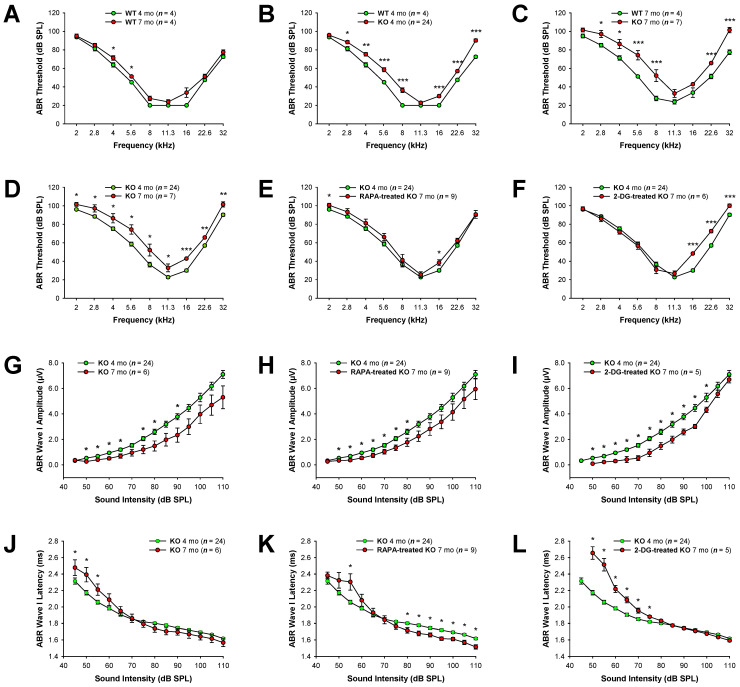
Auditory brainstem responses (ABRs) in rapamycin- and 2-DG-treated Fus1 KO mice. (**A**–**F**) Shown are graphs illustrating the average ABR threshold as a function of stimulus frequency in (**A**) WT mice at 4 and 7 months of age; (**B**) WT and KO mice at 4 months of age; (**C**) WT and KO mice at 7 months of age; (**D**) KO mice at 4 and 7 months of age; (**E**) untreated and rapamycin-treated KO mice; (**F**) untreated and 2-DG-treated KO mice. (**G**–**I**) Shown are I/O function graphs of the average amplitude of ABR wave I as a function of sound intensity at 16 kHz in (**G**) KO mice at 4 and 7 months of age; (**H**) untreated and rapamycin-treated KO mice; (**I**) untreated and 2-DG-treated KO mice. (**J**–**L**) Shown are I/O function graphs of the average latency of ABR wave I as a function of sound intensity at 16 kHz in (**J**) KO mice at 4 and 7 months of age; (**K**) untreated and rapamycin-treated KO mice; (**L**) untreated and 2-DG-treated KO mice. Data are presented as mean ± SEM. * = *p* < 0.05, ** = *p* < 0.01, *** = *p* < 0.001 (Student’s *t*-test).

**Figure 2 antioxidants-12-01225-f002:**
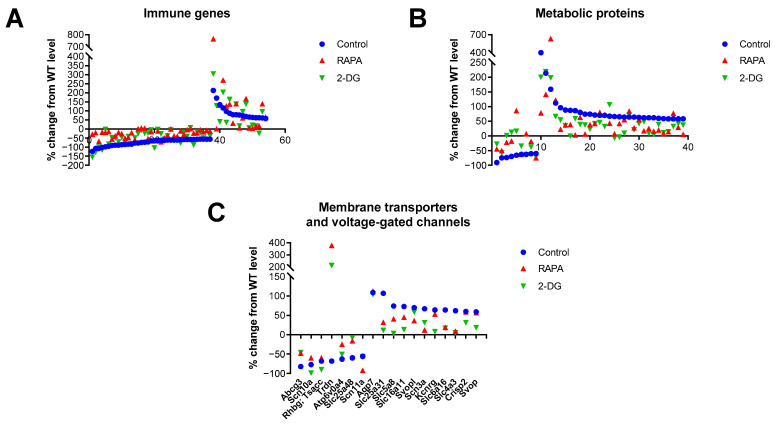
Drug-induced changes in the expression of genes related to immunity, metabolism, and membrane transporters/channels in the cochleae of 7-month-old Fus1 KO mice. (**A**–**C**) These plots illustrate the percentage change in the expression of genes related to (**A**) immunity, (**B**) metabolism, and (**C**) membrane transporters and voltage-gated channels in the cochleae of untreated control and drug-treated Fus1 KO mice relative to WT mice. The x-axis represents different genes that showed altered expression after drug treatment (see Appendix A for the numbered list of genes). The points below the x-axis represent downregulated genes, while the points above the x-axis represent upregulated genes.

**Figure 3 antioxidants-12-01225-f003:**
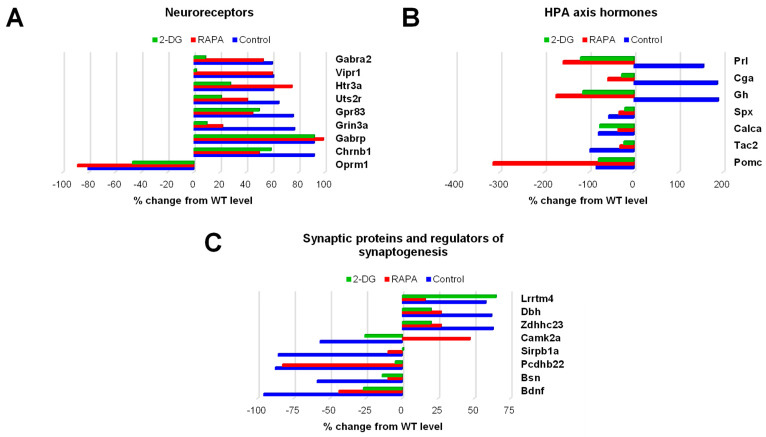
Drug-induced changes in the expression of genes related to neuroreceptors, HPA axis hormones, and synapses/synaptogenesis in the cochleae of 7-month-old Fus1 KO mice. (**A**–**C**) These plots illustrate the percentage change in the expression of genes related to (**A**) neuroreceptors, (**B**) hormones of the hypothalamus–pituitary–adrenal (HPA) axis, and (**C**) synapses and regulators of synaptogenesis in the cochleae of untreated control and drug-treated Fus1 KO mice relative to WT mice. The plots on the right of the axis represent upregulated genes, while the plots on the left of the axis represent downregulated genes.

**Figure 4 antioxidants-12-01225-f004:**
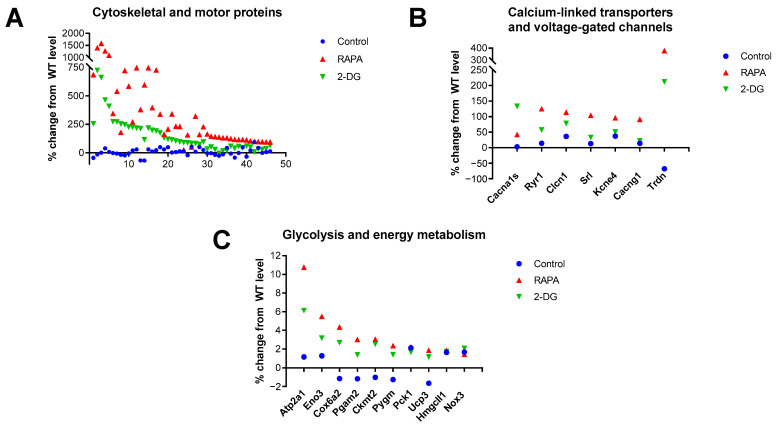
Drug-induced changes in the expression of genes related to the cytoskeleton and motility, calcium-linked transporters/channels, and glycolysis and energy metabolism in the cochleae of 7-month-old Fus1 KO mice. (**A**–**C**) These plots illustrate the percentage change in the expression of genes related to (**A**) cytoskeletal- and motility-associated proteins, (**B**) calcium-linked transporters and voltage-gated channels, and (**C**) glycolysis and energy metabolism in the cochleae of untreated control and drug-treated Fus1 KO mice relative to WT mice. The x-axis represents different genes that showed altered expression after drug treatment (see Appendix A for the numbered list of genes).

**Figure 5 antioxidants-12-01225-f005:**
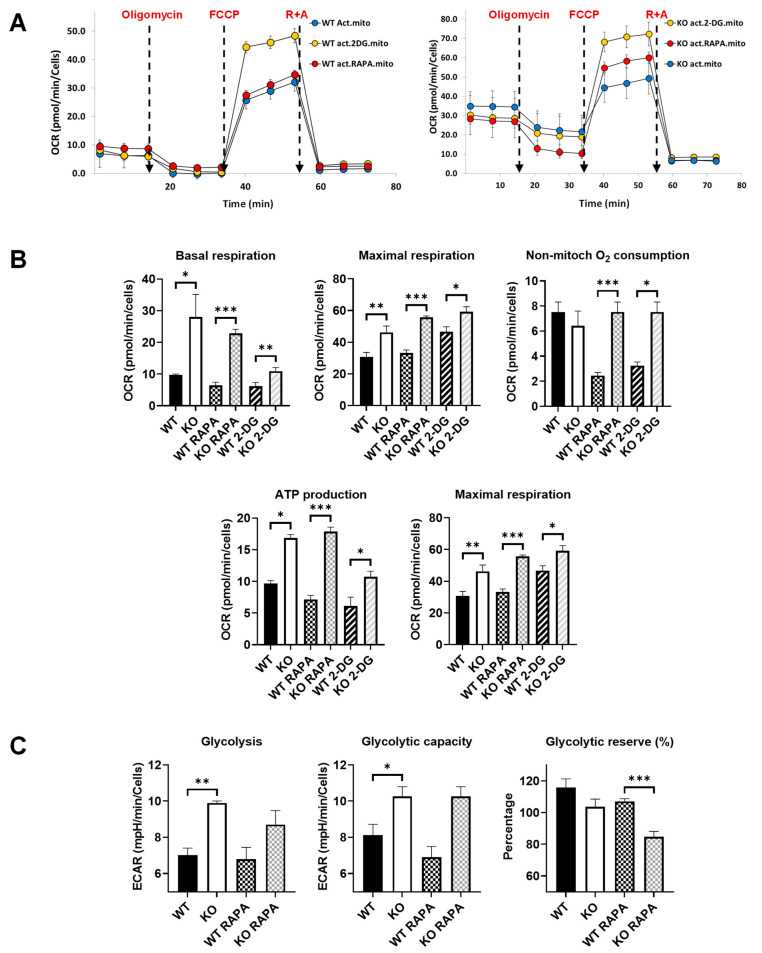
Comparative energetic metabolism of LPS-activated Fus1 KO and WT bone-marrow-derived macrophages (BMDMs) in the absence or presence of RAPA and 2-DG. (**A**) Graphs showing representative Seahorse mitochondrial stress test graphs of WT and KO BMDMs treated or not treated with energy-modifying drugs. Oligomycin (1 µM), FCCP (1 µM), R + A (rotenone + antimycin) (0.5 µM each) were injected at the time shown with the dashed lines. (**B**,**C**) Bar graphs showing quantified protein-normalized mitochondrial parameters calculated based on oxygen consumption rates (OCR) interrogated using the mitochondrial stress test (**B**) and extracellular acidification rates (ECAR) obtained through the glycolysis test (**C**). Glucose dependence was assessed by sequentially injecting oligomycin (1 µM), glucose (10 mM), and 2-deoxyglucose (2-DG, 50 µM). Data (n = 4 separate wells per group) represent 3 experiments. Rapamycin and 2-DG were added along with LPS (100 ng/mL) 4 h prior to performing the tests. OCR and ECAR data were analyzed with Student’s *t*-test. * = *p* < 0.05, ** = *p* < 0.01, *** = *p* < 0.001. Data are presented as mean ± SEM.

**Table 1 antioxidants-12-01225-t001:** Altered biological processes in the Fus1 KO mouse cochlea (KEGG analysis). This table illustrates the number of pathways in different biological categories that were upregulated and downregulated in the cochleae of untreated 7-month-old Fus1 KO vs. WT mice, as revealed by KEGG (Kyoto Encyclopedia of Genes and Genomes) pathway analysis.

Biological Pathway Category	Number of Upregulated Pathways	Number of Downregulated Pathways
Neural response	1	1
Metabolism and associated diseases	9	1
Immune system and associated diseases	1	17
Cell cycle	0	1
Signal transduction	0	2
Cell adhesion	0	1

**Table 2 antioxidants-12-01225-t002:** Rapamycin-induced gene expression changes in the Fus1 KO mouse cochlea (KEGG analysis). This table illustrates changes in the number of pathways in different biological categories that were upregulated and downregulated in the cochleae of untreated vs. rapamycin (RAPA)-treated 7-month-old Fus1 KO mice, as revealed by KEGG (Kyoto Encyclopedia of Genes and Genomes) pathway analysis.

Biological Pathway Category	KO Control ⬆	KO RAPA ⬆	KO Control ⬇	KO RAPA ⬇
Neural response	1	1	1	0
Calcium signaling and associated diseases	0	5	0	0
Metabolism and associated diseases	9	10	1	0
Immune system and associated diseases	1	10	17	0
Cell cycle	0	0	1	0
Signal transduction	0	3	2	0
Cell adhesion	0	4	1	0

Highlighted in grey are pathway categories that were noticeably altered by RAPA treatment. **⬆**= upregulation, **⬇**= downregulation.

**Table 3 antioxidants-12-01225-t003:** 2-DG-induced gene expression changes in the Fus1 KO mouse cochlea (KEGG analysis). This table illustrates changes in the number of pathways in different biological categories that were upregulated and downregulated in the cochleae of untreated vs. 2-deoxy-D-glucose (2-DG)-treated 7-month-old Fus1 KO mice, as revealed by KEGG (Kyoto Encyclopedia of Genes and Genomes) pathway analysis.

Biological Pathway Category	KO Control ⬆	KO 2-DG ⬆	KO Control ⬇	KO 2-DG ⬇
Neural response	1	1	1	1
Metabolism and associated diseases	9	1	1	1
Immune system and associated diseases	1	4	17	8
Cell cycle	0	0	1	1
Signal transduction	0	0	2	1
Cell adhesion	0	1	1	0

Highlighted in grey are pathway categories that were substantially altered by 2-DG treatment. **⬆**= upregulation, **⬇**= downregulation.

**Table 4 antioxidants-12-01225-t004:** Drug-specific correction of critical biological processes dysregulated in the Fus1 KO mouse cochlea. This schematic illustrates biological processes that were upregulated or downregulated in the cochleae of 7-month-old Fus1 KO mice in response to rapamycin (RAPA) and 2-deoxy-D-glucose (2-DG) treatment. **⬆**= upregulation, **⬇**= downregulation.

RAPA	2-DG
Calcium signaling **⬆**	Metabolism **⬇** Immune system **⬆**
Immune system **⬆**	
Signaling transduction **⬆**	
Cell adhesion **⬆**	

## Data Availability

All the data presented in this study are available in the article and Appendix A.

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
