# Peer review of "Pharmacological Modulation of Energy and Metabolic Pathways Protects Hearing in the Fus1/Tusc2 Knockout Model of Mitochondrial Dysfunction and Oxidative Stress"

_antioxidants, 2023, doi:10.3390/antiox12061225_

Round 1

Reviewer 1 Report

In this study, Tan et al. have explored the potential therapeutic benefits of rapamycin and 2-deoxy-glucose in protection against metabolic age-related hearing loss. This is a very interesting study that would be of interest to other researchers in the field. The gene expression and seahorse energetic experiments seem to be well-designed, but the hearing function test is a bit underwhelming due to poor experimental design and statistical analyses (see below). In this reviewer's opinion, a reanalysis of ABR data will significantly improve the manuscript and strengthen the conclusions.

Highlights:

1)      Use of a novel mouse model for ARHL: Fus1/Tusc2 knockout

2)      Rapamycin treatment delayed the progression of hearing loss in Fus1 KO mice. 2DG showed a trend for protection.

3)      Immune and metabolism-related pathways were mainly altered in the Fus1 KO. A dysregulation of HPA axis was also observed in the knockout mice.

4)      Drug treatments reverse gene expression changes and mitochondrial metabolism. 

Weakness:

The study’s main weakness is the lack of clarity in statistical analyses. Were homogeneity of variance and normality tested?

For ABR measurements, it is stated that Student’s t-test was used for the statistical test. This seems to be an inappropriate test. A multivariate ANOVA or a 2-factor ANOVA followed by a post hoc test is more appropriate here. The sample sizes are grossly unequal. Although equal sample size is not an assumption to be satisfied for t-test or ANOVA, having an equal sample size would have strengthened the rigor of the study. 

Reviewer 2 Report

The authors follow up on their former findings on Fus1 KO mice and present evidence that treatment with rapamicin or 2-deoxy-D-glucose protects hearing in these mice and normalizes expression of some of the dysregulated genes, which mainly belong to the mitochondrial metabolism and immune response pathways.

I have appreciated that microarray analysis of differentially regulated genes was followed by comprehensive pathway analysis. I wonder wether some of the most interesting targets that were dysregulated in KO mice and normalized by pharmacological treatment -for example, BSN or BDNF -  have been confirmed by qQPR and/or on a protein level.

The authors should comment on why only female mice have been included in the study and the large difference in mice number between untreated 4 month and treated 7 month mice (see panels D-L of figure 1). Also, 2 mice died during treatment with 2-DG. Can this be attributed to 2-DG toxicity?

I was a bit confused on why authors compared audiological findings of untreated 4 month mice with those of treated 7 month mice. Are the conclusions also supported when comparing treated 7 month mice to untreated 7 month mice?

The authors state that "A 514 constantly high energy production might be detrimental for tissue metabolism and physiologic functioning that could lead to the loss of tissue homeostasis and premature aging. Thus, an attractive strategy for preventing or delaying hearing loss would be the modulation of energy producing and regulating pathways in the cochlea." These conclusions are drawn from the data on activated macrophages. It remains to be established, or at least commented on, to what extent macrophages are representative of the other tissues of the cochlea. Do the authors imply that modulation of the metabolism of macrophages could ameliorate cochlear aging? 

Minor comments:

Please double check table S3A and B: downregulated genes are incorrectly labelled as "upregulated" and vice versa

Line 405: synaptic/synaptogenic proteins (4 upregulated and 4 downregulated), I think 3 are upregulated and 5 downregulated

Results: the authors should comment on the results of the ECAR, especially following pharmacological treatment. The data presented in figure 5c are not mentioned in the main text. 

Author Response

Reviewer #2

We would like to thank Reviewer 2 for their thoughtful comments and suggestions. We appreciate their careful attention to detail, which has helped improved the accuracy of our manuscript.

The authors follow up on their former findings on Fus1 KO mice and present evidence that treatment with rapamicin or 2-deoxy-D-glucose protects hearing in these mice and normalizes expression of some of the dysregulated genes, which mainly belong to the mitochondrial metabolism and immune response pathways.

  • I have appreciated that microarray analysis of differentially regulated genes was followed by comprehensive pathway analysis. I wonder whether some of the most interesting targets that were dysregulated in KO mice and normalized by pharmacological treatment-for example, BSN or BDNF - have been confirmed by qQPR and/or on a protein level.

We appreciate the reviewer’s suggestion to confirm the expression of targets that were dysregulated in the KO cochlea identified in our microarray and pathway analysis using qPCR and/or protein-level analysis. However, we would like to clarify that we did not perform qPCR or protein expression analysis for specific targets in this study. This would require repeating the experiment with larger size groups to have enough cochlear tissues for molecular analysis. The focus of our present study was on identifying dysregulated pathways and potential targets for pharmacological intervention, and we acknowledge that further validation of specific targets would be valuable for future studies.

  • The authors should comment on why only female mice have been included in the study and the large difference in mice number between untreated 4 month and treated 7month mice (see panels D-L of figure 1). Also, 2 mice died during treatment with 2-DG. Can this be attributed to 2-DG toxicity?

Regarding the use of only female mice in our study, we decided to focus on female mice as previous studies have demonstrated sex-specific differences in the efficacy of certain treatments, so we wanted to limit potential confounding factors. We have now added a brief explanation for this choice in the methods section of the manuscript (page 5, line 220).

As mentioned in our response to Reviewer 1, the reason for the larger number of mice in the untreated 4 month-old KO group is that we combined the control mice from different studies, including the rapamycin and 2-DG studies.

Regarding the two Fus1 KO mice that died during 2-DG treatment, while we cannot definitively attribute their deaths to 2-DG toxicity, we acknowledge the possibility that it may have played a role.

  • I was a bit confused on why authors compared audiological findings of untreated 4 month mice with those of treated 7 month mice. Are the conclusions also supported when comparing treated 7 month mice to untreated 7 month mice?

The comparison of ABR findings between untreated 4 month-old and treated 7 month-old Fus1 KO mice was to demonstrate the efficacy of rapamycin and 2-DG treatment in improving the age-related increase in ABR thresholds from 4 to 7 months of age (shown in Fig.1) while both drugs had no effect on ABRs in WT mice. We chose not to present all the comparisons in this manuscript as in our previous study we demonstrated that WT mice do not exhibit significant ABR threshold shifts up to 12-13 months of age, while Fus1 KO mice experience statistically significant loss of hearing at 2-3 months, 4-5 months, 8-10 months, and 12-13 months of age (DOI: 10.1089/ars.2016.6851).

  • The authors state that "A 514 constantly high energy production might be detrimental for tissue metabolism and physiologic functioning that could lead to the loss of tissue homeostasis and premature aging. Thus, an attractive strategy for preventing or delaying hearing loss would be the modulation of energy producing and regulating pathways in the cochlea." These conclusions are drawn from the data on activated macrophages. It remains to be established, or at least commented on, to what extent macrophages are representative of the other tissues of the cochlea. Do the authors imply that modulation of the metabolism of macrophages could ameliorate cochlear aging?

“In our previous study (Tan at al., 2019, ARS) we found that Fus1 KO mice have a higher number of activated cochlear macrophages in the bone marrow of the otic capsule and an increased infiltration of activated macrophages in the spiral ligament, suggesting chronic inflammation in the Fus1 KO cochlea. Activation of immune cells results in profound changes in their energy metabolism, which, if chronic, could be harmful to immune cells and surrounding tissue functions. Thus, we investigated if and how the energy metabolism of activated Fus1 KO and WT macrophages differ and if it could be modulated by RAPA or 2-DG treatments” (cited paragraph is from the section 3.9 of the manuscript, page 13, lines 491-498). Therefore, modulation of macrophage metabolism could potentially ameliorate cochlear aging.

We do realize that cochlear macrophages are functionally different from other cochlear cells, however, considering the systemic loss of Fus1 in these mice, each cochlear cell is affected. The magnitude of the effect depends on the energetic requirement of the cells. Immune, neural, strial, and hair cells have the highest energy consumption among all the cochlear cells and may be affected by Fus1 loss similar to macrophages. Therefore, we used macrophages as a model of cochlear cells with high ATP demands.

Minor comments:

  • Please double check table S3A and B: downregulated genes are incorrectly labelled as "upregulated" and vice versa

We thank the reviewer for bringing this to our attention. We have now made the necessary corrections, as requested.

  • Line 405: synaptic/synaptogenic proteins (4 upregulated and 4 downregulated), I think 3 are upregulated and 5 downregulated

We thank the reviewer for pointing out this error. After re-examining our data, we agree with their observation and have made the necessary corrections.

  • Results: the authors should comment on the results of the ECAR, especially following pharmacological treatment. The data presented in figure 5c are not mentioned in the main text.

We commented on the ECAR results (Fig 5C) in the main text (page 14, lines 511-518).

Reviewer 3 Report

Studies were conducted to continue characterization of a previously established mouse model of metabolic ARHL created by deleting a critical mitochondrial protein, and to determine the effects of oral delivery of RAPA or 2-DG on ARHL in 7-mo female KO mice. Rationale for selection of RAPA and 2-DG was well developed. Results showed interesting changes in pathways linked to immunity (17 pathways downregulated) and high upregulation of cytoskeletal and motor proteins in RAPA and 2-DG-treated cochleae. Overall, the results provide insight into cochlear energy metabolism and mechanisms underlying metabolic presbycusis, and have implications for devising protection strategies. 

The studies are well-designed and appear to have been carefully conducted and clearly reported. I have no major concerns or criticisms, only these minor points to address:

1) L176, How was “significant enrichment” defined and determined?

2) Fig. 1 why are no 4 mo WT amplitude and latency functions shown?

3) L 225, “starting at 4 months of age”: since 4 mo was the lowest age tested, you know that HL was present by 4 mo, but you don’t know that it started at 4 months

4) L279, given the absence of effects of treatment on ABR amplitudes, saying that RAPA and 2-DG “profoundly affect the integrity of the peripheral and central auditory pathways” seems overstated.

Author Response

Reviewer #3

Studies were conducted to continue characterization of a previously established mouse model of metabolic ARHL created by deleting a critical mitochondrial protein, and to determine the effects of oral delivery of RAPA or 2-DG on ARHL in 7-mo female KO mice. Rationale for selection of RAPA and 2-DG was well developed. Results showed interesting changes in pathways linked to immunity (17 pathways downregulated) and high upregulation of cytoskeletal and motor proteins in RAPA and 2-DG-treated cochleae. Overall, the results provide insight into cochlear energy metabolism and mechanisms underlying metabolic presbycusis, and have implications for devising protection strategies.

The studies are well-designed and appear to have been carefully conducted and clearly reported. I have no major concerns or criticisms, only these minor points to address:

We would like to thank Reviewer 3 for their encouraging feedback. We appreciate the reviewer’s positive comments regarding our study and their suggestions to improve to clarity of our manuscript.

1) L176, How was “significant enrichment” defined and determined

For the KEGG pathway analysis that was ran on the gene sets created using fold change x1.3 and expression level intensity 4 (Log2 value) cut off parameters, significant pathway enrichment was considered when adjusted p value was ≤ 0.05. We have now added this brief explanation in the methods section of the manuscript (line 177)

2) Fig. 1 why are no 4 mo WT amplitude and latency functions shown?

Our focus here was to illustrate the age-related changes in ABR wave amplitudes and latencies in untreated and treated Fus1 KO mice. While we did not present the WT data in the present study, we have fully described the hearing phenotype of aging WT and Fus1 KO mice in our previous publication*. Therefore, readers interested in WT data can refer to our previous work. As mentioned in our response to Reviewer 2, both drugs had no significant effect on ABRs in WT mice.

(*Tan, W.J.T., et al., 2017. Novel role of the mitochondrial protein Fus1 in protection from premature hearing loss via regulation of oxidative stress and nutrient and energy sensing pathways in the inner ear. Antioxid. Redox Signal 27(8), 489-509)

3) L 225, “starting at 4 months of age”: since 4 mo was the lowest age tested, you know that HL was present by 4 mo, but you don’t know that it started at 4 months

In our previous study, we reported significant ABR threshold elevations across the frequency range in Fus1 KO mice starting at 4 months of age. To provide additional clarification, we have revised our manuscript to explain why we decided to begin treatment at 4 months of age (line 213)

4) L279, given the absence of effects of treatment on ABR amplitudes, saying that RAPA and 2-DG “profoundly affect the integrity of the peripheral and central auditory pathways” seems overstated.

Yes, we agree with the reviewer that our original statement may have been overstated. To provide a more accurate interpretation of the findings, we have now revised the sentence as follows: “Our analysis of the ABR waveforms reveals that RAPA and 2-DG affects the integrity of the peripheral and central auditory pathways, specifically in relation to ABR wave latencies” (lines 284-286).

Round 2

Reviewer 2 Report

The authors "...acknowledge that further validation of specific targets would be valuable for future studies", but do not mention such a lack of validation in the limitations of the study - which I have recommended to add in the bullet point "Recommendations for Authors", nor mention such a validation as a possible future perspective.

The authors state that "Because previous studies have demonstrated sex-specific differences in the efficacy of certain treatments, we decided to use only female mice to limit potential confounding factors", but do not provide a reference for these previous studies.
